# Blood-, Tissue- and Urine-Based Prognostic Biomarkers of Upper Tract Urothelial Carcinoma

**DOI:** 10.3390/diagnostics14171927

**Published:** 2024-08-31

**Authors:** Jan Łaszkiewicz, Wojciech Krajewski, Aleksandra Sójka, Łukasz Nowak, Joanna Chorbińska, José Daniel Subiela, Wojciech Tomczak, Francesco Del Giudice, Bartosz Małkiewicz, Tomasz Szydełko

**Affiliations:** 1University Center of Excellence in Urology, Wrocław Medical University, 50-556 Wrocław, Poland; 2Department of Minimally Invasive and Robotic Urology, University Center of Excellence in Urology, Wrocław Medical University, Borowska 213, 50-556 Wrocław, Poland; 3Department of Urology, Hospital Universitario Ramón y Cajal, IRYCIS, Universidad de Alcala, 28034 Madrid, Spain; 4Department of Maternal Infant and Urologic Sciences, “Sapienza” University of Rome, Policlinico Umberto I Hospital, 00161 Rome, Italy; 5Department of Urology, Stanford University School of Medicine, Stanford, CA 94305, USA

**Keywords:** upper tract urothelial carcinoma, UTUC, biomarkers, prognosis, survival, oncological outcomes

## Abstract

Upper tract urothelial carcinoma (UTUC) is a rare but aggressive neoplasm. Currently, there are few reliable and widely used prognostic biomarkers of this disease. The purpose of this study was to assess the prognostic value of blood-, tissue- and urine-based biomarkers in patients with UTUC. A comprehensive literature search was conducted using the PubMed, Cochrane and Embase databases. Case reports, editorials and non-peer-reviewed literature were excluded from the analysis. As a result, 94 articles were included in this review. We evaluated the impact of 22 blood-based, 13 tissue-based and 4 urine-based biomarkers and their influence on survival outcomes. The neutrophil–lymphocyte ratio, albumin, C-reactive protein, De Ritis ratio, renal function and fibrinogen, which are currently mentioned in the European Association of Urology (EAU) guidelines, are well researched and most probably allow for a reliable prognosis estimate. However, our review highlights a number of other promising biomarkers that could potentially predict oncological outcomes in patients with UTUC. Nonetheless, the clinical value of some prognostic factors remains uncertain due to the lack of comprehensive studies.

## 1. Introduction

Upper tract urothelial carcinoma (UTUC) is a relatively rare disease accounting for 5–10% of all urothelial carcinomas [1]. It is considered an aggressive neoplasm with high rates of recurrence and progression [2].

Radical nephroureterectomy (RNU) with bladder cuff excision is the standard treatment for UTUC. However, in low-risk cases, kidney-sparing techniques might be employed [3]. Therefore, it is important to identify patients who may benefit from conservative procedures. A variety of clinical, pathological, radiological and tumor characteristics have proven valuable as prognostic factors in UTUC. However, the value of numerous blood-, tumor tissue- and urine-based markers remains unclear [2].

The aim of this review was to summarize the available biomarkers that could be used in the risk stratification of patients with UTUC.

## 2. Materials and Methods

A comprehensive literature search limited to English-language articles published until May 2024 was conducted using the PubMed, Cochrane and Embase electronic databases. Case reports, editorials and non-peer-reviewed literature were excluded from the analysis. The search string included, but was not limited to, the following terms: “upper tract urothelial carcinoma”, “UTUC”, “risk factor”, “biomarker”, “marker”, “prognosis”, “progression”, “prediction”, “recurrence”, “survival outcome” and “oncological outcome”. As a result, 94 articles were included in this review (Figure 1).

## 3. Results

### 3.1. Blood-Based Biomarkers

Because of technical reasons, obtaining representative tumor tissue samples is not always feasible and/or reliable. Therefore, blood-based biomarkers are an attractive alternative. They can be obtained without implementation of invasive procedures, thereby reducing the risk for the patients.

The current European Association of Urology (EAU) guidelines propose using the neutrophil–lymphocyte ratio (NLR), albumin, C-reactive protein (CRP), De Ritis ratio, renal function and fibrinogen as prognostic indicators of tumor progression [2]. These biomarkers, as well as the albumin to globulin ratio (AGR), are the best researched and might be the most valuable in clinical practice. The impact of blood-based biomarkers on patients with UTUC is summarized in Table 1.

#### 3.1.1. C-Reactive Protein

Given the data showing that systemic inflammation takes part in tumor development and progression, CRP, which is an acute-phase protein, may serve as a biomarker for predicting outcomes in cancer patients. Recently, numerous studies have reported that increased CRP levels correlated with poorer prognosis in certain solid tumors [50,51,52,53]. Data regarding the use of CRP in the UTUC prognosis were reported in retrospective studies and meta-analyses. A comprehensive meta-analysis included 10 studies with 2252 patients and demonstrated that an elevated CRP level was strongly associated with worse cancer-specific survival (CSS) in UTUC patients who underwent RNU [4]. Similar results were obtained in another meta-analysis conducted by Zhou et al. [5]. Moreover, a multi-institutional study published by Tanaka et al. reported that increased preoperative CRP levels were associated with worse recurrence-free survival (RFS) and CSS disease recurrence and cancer-specific mortality after RNU [6]. Another study proved that a high pretreatment CRP level was a reliable prognostic factor for a worse overall survival (OS) in clinical node-positive patients who underwent RNU, (neo)adjuvant chemotherapy, radiotherapy and/or palliative care [7]. In addition, Nishikawa et al. selected 135 patients with UTUC who subsequently underwent RNU and showed that CRP was a negative predictor of a higher pathological stage after RNU [8].

#### 3.1.2. Fibrinogen

Increased expression of fibrinogen is associated with malignant cell growth and proliferation; thus, its pretreatment assessment was proposed to predict survival in various tumors [9]. In the literature on UTUC, only retrospective studies and a meta-analysis are available.

A meta-analysis that included five studies and 1050 patients published by Song et al. stated that high pretreatment plasma fibrinogen levels were associated with a worse OS and CSS in UTUC [9]. Other studies suggested that an elevated plasma fibrinogen predicted unfavorable CSS, OS, RFS and progression-free survival (PFS) in UTUC patients treated with RNU [10,11]. Moreover, Tanaka et al. evaluated a cohort of 218 patients and reported that in patients with plasma fibrinogen levels ≥ 450 mg/dL, the 5-year RFS and CSS rates were significantly lower than in patients with plasma fibrinogen levels < 450 mg/dL [54]. The significant association between fibrinogen and CSS after RNU was also supported by Mori et al. [4]. In addition, Xu et al. proved that elevated plasma fibrinogen was strongly correlated with aggressive tumor features, such as a higher tumor stage and grade, nodal involvement, lymphovascular invasion, positive surgical margins, tumor size and sessile architecture [11].

#### 3.1.3. Complete Blood Count (CBC)

The CBC components, including hemoglobin, red blood cells (RBCs), white blood cells (WBCs) and platelets, have been found to reflect systemic inflammation associated with cancer development and progression. The response to cancer, including immune system activation and inflammatory response, is reflected in the CBC parameters, which can help to assess the aggressiveness of the disease and tolerance to treatment. The CBC is routinely tested before the surgery; therefore, it could be easily used to estimate the prognosis in UTUC. In the existing literature, only retrospective studies were found. Cheng et al. analyzed 195 patients with UTUC, who had undergone RNU, and demonstrated that a higher absolute WBC count, red cell distribution width (RDW) and neutrophil to lymphocyte ratio (NLR) were significant predictors of shorter OS. Furthermore, absolute leukocytosis was also a negative prognostic factor for CSS after RNU [12]. Nonetheless, Sheth et al. reported that increased WBCs was more commonly associated with lymphovascular invasion (LVI), but not with RFS and OS [13]. Another research study showed a strong association of elevated WBCs with CSS, in contrast to the platelet (PLT) count, which was not significantly associated with CSS in UTUC [4]. However, Foerster et al. included 2492 patients undergoing RNU and showed that preoperative thrombocytosis predicted the ≥pT2 stage, high tumor grade, lymph node metastasis, LVI, sessile tumor architecture, necrosis and CIS. Furthermore, univariable analyses revealed that elevated platelets were related to worse RFS and OS, although no such correlation was observed in multivariable analyses [14]. On the contrary, the most recent study from 2024 proved that preoperative thrombocytosis was strongly correlated with RFS and CSS in patients treated with RNU. What is more, the authors reported that patients with elevated preoperative platelet levels were at risk of a higher tumor stage, lymph node metastasis, prior bladder cancer diagnosis and preoperative anemia [15]. Additionally, a significant association between preoperative anemia and poor CSS and RFS after RNU has been proven [4,13,15].

#### 3.1.4. Neutrophil to Lymphocyte Ratio

The NLR has been reported to be an important prognostic indicator of cancer progression due to the increase in tumor-associated neutrophils, which directly promote tumor development and low lymphocyte levels, suggesting a weakened immune response against the tumor [55,56]. The prognostic value of the NLR in UTUC patients treated with RNU has been proved in several studies. The current literature mainly consists of meta-analyses and retrospective studies. Three independent meta-analyses revealed that an elevated preoperative NLR was an independent predictor of worse OS, RFS and CSS [16]. Moreover, Zhan et al. suggested that a high NLR was associated with poor OS, while Mori et al. demonstrated its correlation with CSS [4,17]. In addition, Nishikawa et al. retrospectively analyzed a cohort of 135 patients who qualified for RNU and showed that an increased pretreatment NLR had a negative impact on the extravesical RFS and pathological stage [8]. Also, a meta-analysis by Marchioni et al. from 2017 showed that an elevated NLR was significantly associated with OS and RFS but had no influence on CSS [18]. Additionally, Kim et al., in their recent study from 2023, stated that a high postoperative NLR predicted poor OS and CSS after RNU [19].

#### 3.1.5. Systemic Immune–Inflammation Index (SII)

The SII, which is based on the peripheral lymphocyte, neutrophil, and platelet counts, has been recently described as a cancer prognostic biomarker, reflecting the balance between inflammation and immune response [57,58].

Elevated SII values indicate a high inflammatory status, which is known to promote tumor growth and metastasis. The positive correlation between the SII and poor oncologic outcomes was proven in a few studies [57,58,59]. Existing research includes only retrospective studies, which analyzed patients treated with RNU. A study by Mori et al. on 2492 patients showed that a high SII was associated with worse survival outcomes and muscle-invasive disease but not with non-organ-confined disease [20]. Kobayashi et al. had similar results, reporting that an elevated SII was a significant prognostic factor in relation to muscle-invasive disease, as well as poor OS and CSS [21]. Also, two recent studies from 2023 confirmed that an increased SII predicted worse OS [22,23]. Moreover, Jan et al. suggested that a high SII combined with a positive LVI could serve as a predictive marker of poor OS, CSS and PFS [24].

#### 3.1.6. Albumin

Albumin and pre-albumin reflect the nutritional status of patients. Therefore, decreased levels of these markers may predict malnutrition, which is associated with a reduced response to treatment and cancer survival. What is more, a systemic inflammation response to the tumor inhibits albumin synthesis. Thus, hypoalbuminemia was proposed as a predictor of survival in cancer patients. The prognostic value of albumin and pre-albumin in UTUC was described mostly in retrospective studies and one meta-analysis.

A meta-analysis by Liu et al. researched whether serum albumin is suitable for predicting the prognosis in UTUC. The authors found that patients with low preoperative albumin levels had worse OS, CSS and RFS [25]. Moreover, Ku et al. suggested hypoalbuminemia as a negative prognostic marker of worse OS and CSS [26]. On the contrary, a meta-analysis from 2020 presented no significant association of albumin with CSS in UTUC [4]. In addition, Huang et al. retrospectively analyzed 425 patients in research comparing the prognostic value of the preoperative serum pre-albumin and the albumin level. The authors suggested that low preoperative pre-albumin levels, but not albumin, were an independent predictor of reduced patient survival (CSS, OS) [27].

#### 3.1.7. Albumin to Globulin Ratio (AGR)

Albumin and globulin are two key serum proteins, which reflect patients’ nutritional and inflammation status. Thus, it was suggested that the AGR could be used as a prognostic biomarker in various cancers. Most studies in the literature on UTUC are retrospective and only one meta-analysis is available. In a meta-analysis of eight articles from 2022, a low pretreatment AGR was associated with significantly worse OS and CSS in UTUC patients [28]. Similar findings were achieved by Wang et al. [29]. Additionally, a large multicenter study from 2021 reported that patients treated with RNU with a decreased AGR had worse RFS, CSS and OS. A low preoperative AGR was also an independent prognostic factor in muscle-invasive disease and non-organ-confined disease [30]. According to Fukushima et al., a low AGR was an unfavorable factor in terms of the disease-free survival (DFS) and OS after RNU [31]. Furthermore, Xu et al. reported that patients with a low AGR had a greater risk of poor RFS, CSS and OS after RNU. Also, a low AGR was strongly associated with advanced tumor features, including a higher tumor stage and grade, lymph node involvement, LVI, larger tumor size, sessile architecture and concomitant variant histology [32].

#### 3.1.8. Prognostic Nutritional Index (PNI)

The PNI, which is calculated by the serum albumin level and total peripheral lymphocyte count, assesses patients’ nutritional and inflammatory status and risk of postoperative complications. Its prognostic value has been identified in a single meta-analysis by Meng et al., who analyzed six retrospective studies including 2324 UTUC patients. The authors revealed that a low pretreatment PNI was associated with worse OS, CSS, DFS, RFS and PFS after RNU [33].

#### 3.1.9. Lactate Dehydrogenase (LDH)

LDH is the enzyme that converts pyruvate into lactate during glycolysis. The LDH levels increase in tumor cells due to a metabolic shift toward anaerobic glycolysis and adaptation to hypoxic conditions. Furthermore, cancer cells depend on glucose to produce the essential metabolites for their growth, invasion, angiogenesis, and metastasis [60]. These factors contribute to elevated serum LDH, making it a potential prognostic factor in relation to tumor progression. Thus, serum LDH is being used to predict the outcomes and proper patient management in many malignant cancers [61]. The available literature on this biomarker is primarily composed of retrospective studies and one meta-analysis.

In 2020, Wu et al. published a meta-analysis, which included three studies on UTUC and proved that patients with increased pretreatment serum LDH had poor OS and DFS [34]. In a study on 668 patients who underwent RNU, Kaplan–Meier plots showed that preoperative LDH was an independent prognostic factor in terms of unfavorable OS, CSS and RFS, but not of metastasis-free survival (MFS). However, multivariable analysis did not confirm these findings [35]. Sasahara et al. also did not confirm the association between serum LDH and OS [7].

#### 3.1.10. De Ritis Ratio

The De Ritis ratio, which is the ratio between the serum levels of aspartate transaminase (AST) and alanine transaminase (ALT), was initially described to assess liver dysfunction [62]. However, due to the involvement of aminotransferases in cellular metabolism, the AST/ALT ratio was also proposed as a prognostic maker for various malignancies [38,63,64]. Interestingly, the scientific rationale for this phenomenon has not been fully described. The current literature includes meta-analyses and a single observational study. The predictive significance of the pretreatment De Ritis ratio in UTUC was proved in a multi-institutional cohort of 2492 patients undergoing RNU. The authors found that an altered De Ritis ratio was an unfavorable predictor of lymph node metastasis, muscle-invasive disease and non-organ-confined disease. Moreover, it was correlated to high-risk features, such as the LVI, advanced tumor stage, high tumor grade, concomitant CIS and tumor necrosis. Although the De Ritis ratio predicted worse CSS, RFS and OS, it did not show a statistical significance [36]. On the contrary, two independent meta-analyses showed that an elevated AST/ALT ratio was significantly associated with poor CSS, OS, PFS, RFS and MFS in UTUC [37,38].

#### 3.1.11. Alkaline Phosphatase (ALP)

ALP is a key enzyme in bone metabolism. Elevated ALP levels indicate increased osteoblastic activity and bone turnover, which are often associated with bone metastases. Moreover, it was proven that elevated ALP levels were correlated with worse survival outcomes in several malignancies, including prostate, breast and colorectal cancer [65,66,67]. The number of high-quality studies on this subject is limited. Its activity in UTUC was studied by Sheth et al., who demonstrated that ALP levels ≥ 116 IU/L were correlated with aggressive tumor features, such as a higher pathological stage, histologic necrosis and LVI. Also, the authors found that ALP was a negative predictor of RFS and OS in this analysis. Interestingly, they proposed an albumin-to-alkaline phosphatase ratio (AA) in UTUC patients, which proved to be significantly associated with RFS and OS after RNU or ureterectomy [13]. Finally, an analysis of 692 patients from 2018 confirmed that AA was a negative prognostic factor for OS, RFS and CSS [68].

#### 3.1.12. Renal Function Tests

The primary location of UTUC in the kidney may hinder its filtration function [69]. In addition, radical surgery, especially RNU, leads to further renal function loss [70,71,72]. Finally, renal failure is a common complication that excludes patients from chemotherapeutic treatment [72,73]. Therefore, deteriorated kidney function may predict a worse prognosis in patients with UTUC. The available literature on this subject includes meta-analyses and retrospective studies.

Two independent meta-analyses from 2020 and 2021 found that the decline of renal function in patients before RNU was significantly correlated with worse survival rates. It was proven that a low preoperative estimated glomerular filtration rate (eGFR) was an independent predictor of poor OS, PFS and CSS after RNU [4,39]. These results were in agreement with a study from 2022 [40]. Additionally, a large analysis of 731 patients with UTUC showed that an eGFR < 30 mL/min/1.73 m^2^ was an independent risk factor for worse OS and contralateral RFS, but not for CSS. Moreover, severe CKD was related to synchronous contralateral UTUC and concomitant bladder tumor [41]. Moreover, most recent research from 2024 by Muromato et al. revealed that patients with an eGFR < 45 mL/min/1.73 m^2^ presented poorer oncological outcomes in terms of the OS, CSS and non-urothelial tract RFS [42]. In contrast, Nishiwaka et al. claimed that there was no correlation between the eGFR and extravesical RFS [8].

Furthermore, several studies have investigated the predictive value of creatinine in UTUC. Morizane et al. retrospectively analyzed 99 patients and showed that high preoperative serum creatinine levels were significantly associated with worse CSS [43]. However, Mori et al. in their meta-analysis did not confirm these results [4]. In addition, the latest study from 2024 reported that there was no correlation between serum creatinine and OS in 105 UTUC patients who underwent RNU, (neo)adjuvant chemotherapy, radiotherapy and/or palliative care [7].

Cystatin C is a natural cysteine protease inhibitor that is often used to assess renal function. There is growing evidence that cystatin C is involved in the progression of various tumors; nonetheless, its prognostic value in UTUC was not thoroughly investigated [74]. Tan et al. enrolled 538 patients into a study and described that elevated preoperative serum cystatin C correlated with worse survival outcomes in terms of worse CSS, RFS and OS after surgical treatment [44].

#### 3.1.13. Cholinesterases

Cholinesterases are a group of enzymes that hydrolyze the neurotransmitter acetylcholine. The change in the expression of their activities was correlated with worse survival outcomes in urological cancers, including prostate and bladder cancer [75,76]. Nonetheless, there are only two retrospective studies relating to use of this biomarker in UTUC patients who underwent RNU.

Noro et al. proposed butyrylcholinesterase (BChE) as a prognostic indicator in UTUC and showed that increased preoperative serum BChE contributed to longer OS and DFS in patients who underwent RNU [45]. Furthermore, Zhang et al. found a statistically significant correlation between low serum pseudocholinesterase (PChE) and shorter OS and CSS after RNU. Moreover, the authors claimed that decreased PChE levels were significantly associated with a higher tumor stage and muscle-invasive UTUC [46]. In addition, a multi-institutional study from 2023 proved that lower preoperative serum cholinesterase levels predicted advanced tumor features, as well as poor OS, RFS and CSS after RNU [47].

#### 3.1.14. Matrix Metalloproteinases (MMPs)

MMPs are members of a zinc-dependent enzyme family that have been found to be related to tumor progression [77]. Data regarding MMPs in UTUC prognosis is scarce. However, Kovács et al., in the retrospective study, found that increased serum MMPs levels were significantly associated with worse OS and the presence of lymph node or distant metastases in patients who were qualified for RNU, chemotherapy, immunotherapy or a combination of these [48].

#### 3.1.15. Growth Differentiation Factor-15 (GDF-15)

GDF-15, an anti-inflammatory cytokine, has been suggested as a regulator of hepcidin. A retrospective study conducted by Traeger et al. investigated the correlation between these two serum markers (GDF-15 and hepcidin) and UTUC patients’ survival. According to them, higher hepcidin and GDF-15 levels contributed to metastases, tumor recurrence and shorter OS [49]. However, no other study confirmed these results.

### 3.2. Tumor Tissue-Based Biomarkers

Tissue biomarkers require a representative amount of reliable tissue for evaluation. Therefore, it is not always possible to use these parameters when qualifying patients for treatment. However, some of them have proven to have strong predictive values (Table 2). E-cadherin, Ki-67 and p-53 have been widely studied and are probably the most useful in clinical practice.

#### 3.2.1. E-Cadherin

Cadherins are transmembrane glycoproteins that mediate cell–cell adhesion and play an essential role in tissue homeostasis. Loss of E-cadherin expression results in reduced cellular adhesion in epithelial tissues, which is further correlated with invasive growth and metastasis [101]. Hence, the hypothesis of the influence of decreased E-cadherin expression on the UTUC progression was put forward [101,102]. The current literature on this subject includes several retrospective studies and one meta-analysis. A multi-institutional study on 678 patients, who underwent RNU, showed that decreased E-cadherin expression in tumor cells was strongly associated with an advanced tumor stage, high tumor grade, lymph node metastases, CIS and multifocality. Even though E-cadherin failed to be an independent predictor of CSS and RFS, it was correlated with worse outcomes after RNU [78]. A study published in 2002 reported that positive E-cadherin staining was an independent prognostic factor for OS and CSS after nephroureterectomy [79]. Additionally, Reis et al. reported that overexpression of E-cadherin was related to tumor recurrence and shorter time to tumor recurrence in patients who underwent nephroureterectomy or ureterectomy [80]. However, a meta-analysis from 2019 involving 1014 patients found no correlation between E-cadherin and UTUC prognosis [81]. Also, Missaoui et al. reported no significant prognostic value for E-cadherin expression [86].

#### 3.2.2. Ki-67

The Ki-67 antigen is a nuclear protein that is associated with cell proliferation and is widely used as a prognostic marker of malignant tumors. The literature on this biomarker consists mostly of retrospective studies, one prospective study and a meta-analysis. All the following results refer to patients who underwent RNU. In a prospective analysis on 101 patients who underwent RNU or ureterectomy, Ki-67 overexpression predicted an advanced tumor stage, sessile architecture, LVI and non-organ-confined disease [82]. In 2018, Ahn et al. performed a meta-analysis to evaluate the predictive role of Ki-67 in cases with UTUC. After analyzing 12 studies, they reported that increased Ki-67 expression was strongly associated with worse DFS, CSS and OS after RNU [83]. Similarly, a meta-analysis conducted by Fan et al. showed that Ki-67 overexpression was an independent prognostic factor for poor CSS, DFS and MFS, but not for OS and RFS, in patients treated with RNU [84]. Moreover, two reports demonstrated that abnormal Ki-67 expression predicted worse RFS in UTUC patients [82,85]. Nonetheless, Missaoui et al. found no such correlation [86].

#### 3.2.3. p53

The p53 is the key tumor suppressor and a main mediator of cell cycle progression, DNA repair and apoptosis. Mutations in the p53 gene have been identified in various human cancers and correlated with its prognosis. Two meta-analyses and one retrospective research study on p53 prognostic value in UTUC was found in the literature. A meta-analysis of seven articles published in 2013 suggested that positive p53 expression was a potential prognostic marker in UTUC patients qualified for RNU and/or chemotherapy, as it was strongly associated with the DFS, CSS and OS [87]. Another meta-analysis gathering nine studies revealed significant correlation between the p53 expression and the tumor stage and grade [103]. Furthermore, Missaoui et al. found that p53 overexpression predicted an advanced tumor stage, positive surgical margin and shorter RFS [86].

#### 3.2.4. Murine Double Minute 2 (MDM2)

MDM2, which is a negative regulator of p53, promotes cancer cell survival and growth by suppressing the p53-dependent cell cycle and apoptosis. However, the clinical value of MDM2 has not been intensively investigated in UTUC and only one retrospective study can be found on this subject. Bao et al. retrospectively examined 341 UTUC patients treated with RNU and showed that MDM2 was an independent predictor of worse CSS and DFS. Furthermore, increased MDM2 expression was associated with an advanced pathological tumor stage, high tumor grade, ureter localization and presence of glandular and sarcoma differentiation [88].

#### 3.2.5. The Urokinase-Type Plasminogen Activator (uPA) System

The uPA system, comprising the uPA, uPA receptor (uPAR), and uPA inhibitors (PAI-1 and PAI-2), takes part in extracellular matrix regulation, angiogenesis and cell proliferation [104]. What is more, the uPA system was found to be correlated with poor outcomes in cancer patients [104,105].

The potential value of uPA, uPAR and PAI-1 as prognostic markers was described in a single, multicenter research study on 732 patients who underwent RNU. Abufaraj et al. reported that overexpression of all the tested uPA system components was associated with an advanced pathologic tumor stage and LVI. Additionally, uPAR expression independently predicted sessile architecture and tumor necrosis, while PAI-1 expression was found to correlate with multifocal disease. In survival analyses, uPA, uPAR and uPAI-1 expression was a significant negative prognostic factor in univariate analysis but did not have an impact on survival in the multivariate analysis. However, in patients with organ-confined disease (≤pT2N0), uPA was significantly associated with shorter RFS, OS and CSS [89].

#### 3.2.6. SRY-Related HMG-Box 2 (SOX2)

SOX2, which is a member of the SOX family, is a transcription factor that plays a crucial role in embryonic development and tumorigenesis. What is interesting, it that it was also proven that SOX2 was involved in the resistance of cancer cells to different anticancer therapies, such as chemotherapy, radiotherapy and targeted therapy [106]. Therefore, it was hypothesized that SOX2 may predict survival outcomes in UTUC patients. Nonetheless, there is only one retrospective study available in the literature. Bao et al. investigated the predictive value of SOX2 expression by analyzing tissue samples from 341 UTUC patients treated with RNU. The authors revealed that SOX2 overexpression was associated with aggressive pathological features, including a high tumor stage and grade, sessile architecture and the presence of glandular differentiation. In addition, high SOX2 expression was proved to be correlated with poor CSS and DFS [90].

#### 3.2.7. The BRCA1-Associated Protein-1 (BAP1)

BAP1 is a tumor suppressor that regulates the BRCA1 growth control pathway [107]. BAP1 expression was found to be a prognostic factor in several malignancies; however, data on its significance in UTUC is unclear, as there is only one retrospective study available in the literature. In 2018, Aydin et al. gathered a multicenter cohort of 348 UTUC patients who underwent RNU. They showed that BAP1 loss was associated with favorable pathological tumor features, such as the papillary architecture and absence of tumor necrosis and concomitant CIS. Moreover, patients with BAP1 loss had improved RFS and CSS, but not OS [91].

#### 3.2.8. Programmed Death-Ligand 1 (PD-L1)

PD-L1, together with programmed cell death-1 (PD-1), plays a crucial role in suppressing immune cell proliferation and release of immune factors. Apart from that, it has been reported that PD-L1 expression revealed prognostic value in a variety of tumors [108,109,110]. The literature on this potential biomarker consists of retrospective studies and one meta-analysis.

According to a meta-analysis comprising eight studies, overexpression of PD-L1 was associated with shorter CSS, but not with OS, in UTUC patients. Moreover, high PD-L1 expression was an unfavorable predictor of the tumor grade and depth of invasion (pT3 + pT4 + pT2 vs. pT1 + pTa/pTis) [92]. Additionally, a multicenter study on 283 individuals proved that patients with positive PD-L1 expression had ≥pT2 stage, higher tumor grade and LVI. Furthermore, PD-L1 expression correlated with decreased CSS, RFS and OS [93]. Other studies reported that PD-L1-positive UTUC was associated with a higher tumor stage, lymph node invasion, and shorter OS and CSS in patients who underwent RNU, nephroureterectomy and/or ureterectomy [94,111].

#### 3.2.9. Human Epidermal Growth Factor Receptor (HER-2)

HER-2 gene amplification has been found to be involved in tumor proliferation, differentiation and angiogenesis. Moreover, numerous studies have shown it to be correlated with tumor progression and reduced survival rates in different human cancers [112,113,114,115]. In the literature, there are mainly retrospective studies available. Even though HER-2 expression in UTUC cases is rare, several researchers proved that it is associated with worse outcomes. A large multicenter study from 2016 reported that HER-2 overexpression was found in 35.8% of UTUC patients undergoing RNU, who had an increased risk of an advanced tumor stage, high tumor grade and LVI. High HER-2 expression was also a prognostic factor for CSS, RFS and OS [95]. Furthermore, Vershasselt-Crinquette et al. reported strong association between HER-2 overexpression and the pN+ stage, but no correlation with specific survival or recurrence [96].

#### 3.2.10. Enhancer of Zeste Homolog 2 (EZH2)

EZH2 is a histone methyltransferase involved in silencing tumor suppressor genes. Its overexpression has been found in cancer tissues and is correlated with poor clinical outcomes [116,117]. The relationship between EZH2 and UTUC was investigated in a single retrospective multi-institutional study on 376 patients undergoing RNU. The authors revealed that abnormal EZH2 expression was more often associated with ureteral location, sessile architecture, tumor necrosis and concomitant CIS. Univariable analysis showed that EZH2 was significantly associated with worse RFS, CSS and OS; however, multivariable analysis did not confirm statistical significance for RFS [97].

#### 3.2.11. Matrix Metalloproteinase 11 (MMP-11)

MMP-11, a member of the MMP family, plays a role in the degradation of the extracellular matrix. Several studies have reported that MMP11 participated in cancer development by inhibiting apoptosis and promoting migration, invasion and metastasis of tumor cells [118,119,120]. However, there is only one retrospective study regarding the use of MMP-11 in UTUC cases. From a dataset of 340 patients, Li et al. proved that MMP-11 overexpression was an independent negative prognostic factor for CSS and MFS. In addition, increased MMP-11 expression was correlated with the pathologic tumor stage, lymph nodes metastasis, vascular and perineural invasion [98].

#### 3.2.12. Insulin-like Growth Factor Messenger RNA-Binding Protein 3 (IMP3)

IMP3, which is involved in the early stages of embryogenesis, was found to be a biomarker of various tumors [121,122,123]. A single retrospective study from 2013 on 622 UTUC patients who underwent RNU proved that IMP3-positive patients had a higher tumor stage and grade, lymph node involvement, concomitant CIS and LVI. Moreover, multivariable analysis showed a significant association between the IMP3 expression and the CSS, OS, RFS, cancer-specific mortality and disease recurrence after RNU [99].

#### 3.2.13. Pyruvate Dehydrogenase Kinase 3 (PDK3)

PDK3 is one of the main regulatory enzymes of glucose metabolism, which has been identified as a potential regulator of tumorigenesis [124]. The prognostic significance of PDK3 in UTUC was examined by Kuo et al. in a single retrospective study on 340 cases. PDK3 overexpression was proved to be strongly correlated with a high tumor stage and grade, vascular invasion, renal pelvis tumors and high mitotic rate. Moreover, high PKD3 expression was an independent predictor of poor CSS, DFS and MFS. However, no other study has investigated this correlation [100].

### 3.3. Urine-Based Biomarkers

Urine is a readily available biological material that can be obtained non-invasively. Therefore, it is routinely used in the diagnostic evaluation and follow-up of patients with UTUC. Urinary cytology is probably the most crucial biomarker, as it allows for risk stratification in accordance with the EAU guidelines. Various urine-based prognostic biomarkers are presented in Table 3.

#### 3.3.1. DNA Methylation

An increasing interest in the involvement of epigenetic mechanisms in tumorigenesis identified DNA methylation as a diagnostic tool and prognostic indicator in various cancers. To date, a single meta-analysis and two retrospective studies have been conducted on its potential prognostic value in UTUC. Lin et al. analyzed 11 articles reporting 12 methylated genes (RASSF1, GDF15, VIM, HSPA2, TMEFF2, BRCA1, THBS1, ABCC6, SALL3, CDH1, SPARCL1, ONECUT2). Single methylated genes showed different, sometimes contradictory results in terms of the oncological outcomes. Nevertheless, pooled analysis of all the genes has shown that methylation was associated with a lower risk of tumor recurrence but a high risk of progression and mortality [125]. In addition, Monteiro-Reis et al. found that low VIM methylation levels were significantly associated with poor OS and DFS [126].

#### 3.3.2. Fluorescence In Situ Hybridization (FISH)

FISH, which detects chromosomal abnormalities in urine specimens, is a widely used method of cancer diagnosis. Moreover, there are indications that positive FISH may predict worse prognosis in UTUC. However, data on this subject are scarce.

A single retrospective study by Guan et al. researched whether positive FISH is suitable for predicting the prognosis in UTUC patients. They reported that patients with positive FISH results were more likely to present with bladder recurrence. However, no correlation between positive FISH and CSS was found [127].

#### 3.3.3. B2-Microglobulin (B2-MG)

B2-MG is a component of the major histocompatibility complex (MHC) class I molecules, which are present on the surface of almost all nucleated cells. It has been reported that B2-MG activates various signaling pathways that regulate the proliferation, invasion, migration and metastasis of cancer cells [130]. Therefore, B2-MG was suggested to be an independent prognostic factor in cancer patients [131,132,133]. However, it is worth noting that elevated B2-MG levels were also found in renal failure and autoimmune and infectious diseases [130,134,135].

Han et al. suggested that urinary B2-MG could serve as a prognostic biomarker of UTUC. According to them, patients with a high urine B2-MG level had a higher pathologic T stage, large tumor size, perineural invasion and decreased renal function. Furthermore, Kaplan–Meier analysis showed that high B2-MG was associated with significantly worse DFS and MFS, although multivariable Cox regression analysis did not validate these results [128]. Unfortunately, no other research on this topic is available.

#### 3.3.4. Urine Cytology

Urine cytology is a routinely used laboratory technique in UTUC diagnosis. It allows for non-invasive detection of malignant cells, indicating the presence and severity of the disease. Its prognostic significance in UTUC was studied in a retrospective cohort study and a meta-analysis conducted by Fan et al. After analyzing urine samples from 231 patients treated with RNU, the authors concluded that positive preoperative urine cytology was significantly correlated with poor intravesical RFS. Moreover, a meta-analysis revealed that preoperative positive urine cytology was associated with a 49% increased risk of intravesical recurrence [129].

## 4. Conclusions

In summary, our review highlights a number of promising biomarkers that could potentially predict the outcomes in patients with UTUC. However, the clinical value of the available prognosticators remains uncertain due to the lack of comprehensive studies. Therefore, further research is essential in order to validate their utility and improve UTUC patient management.

## Figures and Tables

**Figure 1 diagnostics-14-01927-f001:**
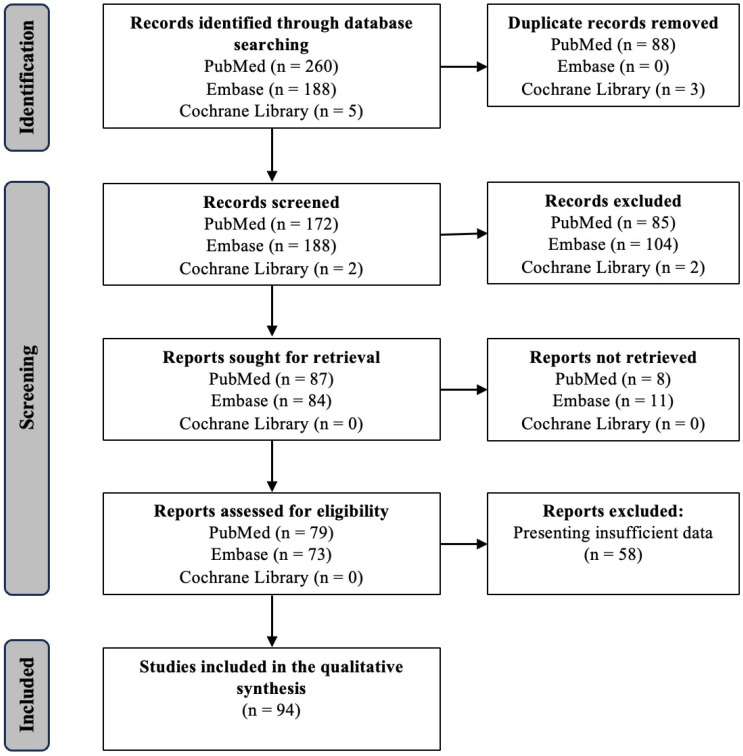
Flow diagram of the study selection process.

**Table 1 diagnostics-14-01927-t001:** Blood-based biomarkers and their impact on the prognosis in patients with upper tract urothelial carcinoma.

Biomarker	First Author	Year	OS	CSS	RFS	Other Survival Outcomes
High CRP	Mori [4]	2020	Worse	Worse	NR	NR
	Zhou [5]	2015	Worse	Worse	NR	NR
	Tanaka [6]	2014	NR	Worse	Worse	NR
	Sasahara [7]	2024	Worse	NR	NR	NR
	Nishikawa [8]	2018	NR	NR	No correlation with extravesical RFS	NR
High Fibrinogen	Song [9]	2019	Worse	Worse	NR	NR
	Liu [10]	2019	NR	Worse	NR	Worse PFS
	Xu [11]	2020	Worse	Worse	Worse	NR
High WBC	Cheng [12]	2015	Worse	Worse	NR	NR
	Sheth [13]	2016	No correlation	NR	No correlation	NR
	Mori [4]	2020	NR	Worse	NR	NR
High RDW	Cheng [12]	2015	Worse	No correlation	NR	NR
High PLT	Foerster [14]	2017	Worse in univariable analysis; no correlation in multivariable analysis	No correlation	Worse in univariable analysis; no correlation in multivariable analysis	NR
	Milojevic [15]	2024	NR	Worse	Worse	NR
Low HGB	Milojevic [15]	2024	NR	Worse	Worse	NR
	Mori [4]	2020	NR	Worse	NR	NR
	Sheth [13]	2016	No correlation	NR	Worse	NR
High NLR	Vartolomei [16]	2018	Worse	Worse	Worse	NR
	Mori [4]	2020	NR	Worse	NR	NR
	Wang [17]	2020	Worse	Worse	NR	NR
	Nishikawa [8]	2018	NR	NR	Worse extravesical RFS	NR
	Marchioni [18]	2016	Worse	No correlation	No correlation	NR
	Kim [19]	2023	Worse	Worse	NR	NR
High SII	Mori [20]	2021	Worse	Worse	Worse	NR
	Kobayashi [21]	2021	Worse	Worse	NR	NR
	Liu [22]	2023	Worse	NR	NR	NR
	Luo [23]	2023	Worse	NR	NR	NR
	Jan [24]	2022	Worse	Worse	NR	Worse PFS
Low albumin	Liu [25]	2018	Worse	Worse	Worse	NR
	Ku [26]	2014	Worse	Worse	NR	NR
	Mori [4]	2020	NR	No correlation	NR	NR
Low pre-albumin	Huang [27]	2017	Worse	Worse	NR	NR
Low AGR	Xia [28]	2022	Worse	Worse	NR	NR
	Zhang [29]	2015	Worse	Worse	NR	NR
	Miura [30]	2021	Worse	Worse	Worse	NR
	Fukushima [31]	2018	Worse	NR	NR	Worse DFS
	Xu [32]	2018	Worse	Worse	Worse	NR
Low PNI	Meng [33]	2022	Worse	Worse	Worse	Worse DFS, PFS
High LDH	Wu [34]	2020	Worse	NR	NR	Worse DFS
	Tan [35]	2018	Worse in Kaplan–Meier analysis; no correlation in multivariable analysis	Worse in Kaplan–Meier analysis; no correlation in multivariable analysis	Worse in Kaplan–Meier analysis; no correlation in multivariable analysis	No correlation with MFS
	Sasahara [7]	2024	Worse	NR	NR	NR
High De Ritis Ratio	Mori [36]	2020	Worse in Kaplan–Meier analysis and univariable analysis; no correlation in multivariable influence	Worse in Kaplan–Meier analysis and univariable analysis; no correlation in multivariable influence	Worse in Kaplan–Meier analysis and univariable analysis; no correlation in multivariable influence	No correlation with MFS in Kaplan–Meier analysis and univariable analysis
	Hu [37]	2020	Worse	Worse	Worse	Worse PFS, MFS
	Su [38]	2020	Worse	Worse	Worse BRFS	Worse PFS
High ALP	Sheth [13]	2016	Worse	NR	Worse	NR
Low eGFR	Kim [39]	2021	Worse	Worse	NR	Worse PFS
	Mori [4]	2020	NR	Worse	NR	NR
	Li [40]	2022	Worse	Worse	NR	Worse PFS
	Bao [41]	2019	Worse	No correlation	Worse	NR
	Muramoto [42]	2024	Worse	Worse	Worse non-urothelial tract RFS	NR
	Nishikawa [8]	2018	NR	NR	No correlation with eRFS	NR
High creatinine	Morizane [43]	2012	NR	Worse	NR	NR
	Mori [4]	2020	NR	No correlation	NR	NR
	Sasahara [7]	2024	No correlation	NR	NR	NR
High cystatin C	Tan [44]	2019	Worse	Worse	Worse	NR
High BChE	Noro [45]	2017	Improved	NR	NR	Improved DFS
Low PChE	Zhang [46]	2016	Worse	Worse	NR	NR
	Von Deimling [47]	2023	Worse	Worse	Worse	NR
High MMPs	Kovács [48]	2022	Worse	NR	NR	NR
High GDF-15	Traeger [49]	2019	Worse	NR	NR	NR

AGR, albumin–globulin ratio; ALP, alkaline phosphatase; BChE, butyrylcholinesterase; BRFS, biochemical recurrence-free survival; CRP, C-reactive protein; CSS, cancer-specific survival; DFS, disease-free survival; eGFR, estimated glomerular filtration rate; GDF-15, growth differentiation factor-15; HGB, hemoglobin; LDH, lactate dehydrogenase; MMPs, metalloproteinases; NLR, neutrophil-lymphocyte ratio; OS, overall survival; PChE, pseudocholinesterase; PFS, progression-free survival; PLT, platelets; RFS, recurrence-free survival; RDW, red cell distribution width; SII, systemic immune-inflammation index; WBC, white blood cell.

**Table 2 diagnostics-14-01927-t002:** Tissue-based biomarkers and their impact on prognosis in patients with upper tract urothelial carcinoma.

Biomarker	First Author	Year	OS	CSS	RFS	Other Survival Outcomes
E-cadherin overexpression	Favaretto [78]	2016	NR	Worse in univariable analysis; no correlation in multivariable analysis	Worse in univariable analysis; no correlation in multivariable analysis	NR
	Fromont [79]	2002	Worse	NR	NR	Worse DFS
	Reis [80]	2012	NR	NR	Worse	NR
	Tae [81]	2019	No correlation	No correlation	No correlation	NR
Ki-67 overexpression	Krabbe [82]	2014	NR	Worse in Kaplan–Meier and univariable analysis; no correlation in multivariable analysis	Worse	NR
	Ahn [83]	2018	Worse	Worse	NR	Worse DFS
	Fan [84]	2016	No correlation	Worse	No correlation	Worse DFS, MFS
	Yang [85]	2022	NR	NR	Worse	NR
	Missaoui [86]	2020	NR	NR	No correlation	NR
p53 overexpression	Ku [87]	2013	Worse	Worse	NR	Worse DFS
	Missaoui [86]	2020	NR	NR	Worse	NR
MDM2 overexpression	Bao [88]	2019	NR	Worse	NR	Worse DFS
uPa system overexpression	Abufaraj [89]	2020	Worse in univariable analysis; no correlation in multivariable analysis Worse in patients with organ-confined disease (≤pT2N0)	Worse in univariable analysis; no correlation in multivariable analysis Worse in patients with organ-confined disease (≤pT2N0)	Worse in univariable analysis; no correlation in multivariable analysis Worse in patients with organ-confined disease (≤pT2N0)	NR
SOX2 overexpression	Bao [90]	2019	NR	Worse	NR	Worse DFS
BAP1 loss	Aydin [91]	2019	No correlation	Improved	Improved	NR
PD-L1 overexpression	Lu [92]	2020	No correlation	Worse	NR	NR
	Campedel [93]	2023	Worse	Worse	Worse	NR
	Chen [94]	2021	Worse	Worse	NR	NR
HER-2 overexpression	Soria [95]	2016	Worse	Worse	Worse	NR
	Vershasselt-Crinquette [96]	2012	Worse	NR	NR	Worse DFS
EZH2 overexpression	Singla [97]	2018	Worse in univariable analysis No correlation with in multivariable analysis	Worse in univariable analysis No correlation with in multivariable analysis	Worse in univariable analysis No correlation with in multivariable analysis	NR
MMP-11 overexpression	Li [98]	2016	NR	Worse	NR	Worse MFS
IMP3 overexpression	Lee [99]	2013	Worse	Worse	Worse	Worse cancer specific mortality, disease recurrence
PDK3 overexpression	Kuo [100]	2021	NR	Worse	NR	Worse MFS

BAP1, BRCA1-associated protein-1; CSS, cancer-specific survival; DFS, disease-free survival; EZH2, enhancer of zeste homolog 2; HER-2, human epidermal growth factor receptor 2; IMP3, insulin-like growth factor messenger RNA-binding protein 3; MDM2, murine double minute 2; MFS, metastasis-free survival; MMP-11, matrix metalloproteinase 11; OS, overall survival; PD-L1, programmed death-ligand 1; PDK3, pyruvate dehydrogenase kinase 3; RFS, recurrence-free survival; SOX2, SRY-related HMG-box 2; uPA, urokinase-type plasminogen activator system.

**Table 3 diagnostics-14-01927-t003:** Urine-based biomarkers and their impact on the prognosis in patients with upper tract urothelial carcinoma.

Biomarker	First Author	Year	OS	CSS	RFS	Other Survival Outcomes
DNA methylation	Lin [125]	2023	NR	NR	Low risk of tumor recurrence	High risk of progression and mortality
	Monteiro-Reis [126]	2013	Worse	NR	NR	Worse DFS
FISH	Guan [127]	2018	NR	No correlation	More frequent bladder recurrence	NR
High B2-MG	Han [128]	2022	NR	NR	NR	Worse DFS and MFS in Kaplan–Meier analysis No correlation with DFS, MFS in multivariable analysis
Urinary cytology	Fan [129]	2021	NR	NR	Worse	NR

B2-MG, B2-microglobulin; CSS, cancer-specific survival; DFS, disease-free survival; MFS, metastasis-free survival; OS, overall survival; RFS, recurrence-free survival.

## Data Availability

No new data were created.

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
