# Peer review of "Blood-, Tissue- and Urine-Based Prognostic Biomarkers of Upper Tract Urothelial Carcinoma"

_diagnostics, 2024, doi:10.3390/diagnostics14171927_

Round 1

Reviewer 1 Report

Comments and Suggestions for Authors

Using literature search, the authors summarized the biomarkers for UTUC prognosis. It is just a simple list of biomarker from published articles. There is no significant contribution to either research filed or clinical use. I would suggest the authors do some further investigation to pick up more valuable biomarkers from the whole list and provide evidence to show the importance. 

1. which biomarker(s) is specific to UTUC?

2. Each of the biomarker showed potential predictive value of UTUC, could the author provide more comprehensive information to show which of these biomarkers are more important.

3. the authors summarized biomarkers from blood, tissue and urine, which way is more reliable? which biomarkers could be found in a kind of universal way? What biomarkers would the authors suggest to be used in clinical space?

Author Response

Comments 1: Using literature search, the authors summarized the biomarkers for UTUC prognosis. It is just a simple list of biomarkers from published articles. There is no significant contribution to either research field or clinical use.

Response: Thank you for taking the time to review our article. We would like to emphasize that our aim was to summarize the available literature on UTUC prognostic biomarkers. In our opinion a significant number of prognosticators were described in the literature and it might be troublesome to find all of the biomarkers in the original articles. Therefore, this review could facilitate the work of clinicians looking for a comprehensive list of the potential prognostic biomarkers of UTUC. As this is a review, no new data have been produced.

Comments 2: I would suggest the authors do some further investigation to pick up more valuable biomarkers from the whole list and provide evidence to show the importance.

Response: In our opinion the best researched, thus the most valuable blood-based prognostic biomarkers are the ones mentioned in the current EAU guidelines: NLR, albumin, CRP, modified Glasgow score, De Ritis ratio, renal function and fibrinogen. In addition, albumin/globulin ratio has been studied quite extensively and has a well-proven prognostic value.

Considering tissue-based biomarkers, e-cadherin, Ki-67 and p-53 have been described as valid prognosticators in many studies.

Finally, urinary cytology is routinely used in risk assessment of UTUC and we think that it still is one of the most important prognostic biomarkers.

We changed the introductions of each of the sections (blood-, tissue- and urine-based biomarkers) in order to highlight the most important prognosticators: 

-“The current European Association of Urology (EAU) guidelines propose using the neutrophil-lymphocyte ratio (NLR), albumin, C-reactive protein (CRP), De Ritis ratio, renal function and fibrinogen as prognostic indicators of tumour progression [2]. These biomarkers, as well as albumin to globulin ratio (AGR) are the best-researched and might be the most valuable in clinical practice” (line 62); 

-”Tissue biomarkers require a representative amount of reliable tissue for evaluation. Therefore, it is not always possible to use these parameters when qualifying for the treatment. However, some of them have proven strong predictive values (Table 2).  E-cadherin, Ki-67 and p-53 have been widely studied and are probably the most useful in clinical practice.” (line 327);

-”Urine is a readily available biological material that can be obtained non-invasively. Therefore, it is routinely used in diagnostic evaluation and follow-up of patients with UTUC. Urinary cytology is probably the most crucial biomarker, as it allows for risk stratification in accordance with the EAU guidelines.” (line 502); 

Comments 3: 1. which biomarker(s) is specific to UTUC?

Response: Unfortunately, none of the prognostic biomarkers included in our review is specific to UTUC. However, we would like to underline that we assessed only the prognostic effect of the markers in patients that were already diagnosed with UTUC.

Comments 4: 2. Each of the biomarker showed potential predictive value of UTUC, could the author provide more comprehensive information to show which of these biomarkers are more important.

Response: Thank you for this comment. This issue has been addressed above.

Comments 5: 3. the authors summarized biomarkers from blood, tissue and urine, which way is more reliable? which biomarkers could be found in a kind of universal way? What biomarkers would the authors suggest to be used in clinical space?

Response: Above we have mentioned the prognostic biomarkers that we think are the most important and could be used in clinical practice. In our opinion, all types of biomarkers (blood-, tissue- and urine-based) can be obtained during the usual diagnostic/therapeutic process and all of them have an important role in the prognosis assessment. However, all of them have limitations. That is why we think that the implementation of different types of biomarkers might provide a more accurate prognosis estimation.

Reviewer 2 Report

Comments and Suggestions for Authors

This manuscript presents a concise overview of potential prognosticators for Upper Tract Urothelial Carcinoma patients. This synopsis of scientific data is very useful and may constitute the basis for future research. The references are numerous and may be very helpful for future investigators. Acceptable for publication.

Author Response

Comments: This manuscript presents a concise overview of potential prognosticators for Upper Tract Urothelial Carcinoma patients. This synopsis of scientific data is very useful and may constitute the basis for future research. The references are numerous and may be very helpful for future investigators. Acceptable for publication.

Response: We would like to thank the Reviewer for the appreciation of our work and for accepting it for publication.

Reviewer 3 Report

Comments and Suggestions for Authors

The authors explored the role of biomarkers(blood, tumor-derived, and urine-based) in upper tract urothelial carcinoma (UTUC). Specifically, they relied on a systematic review methodology including 94 studies. The work is clearly presented, well-read, and comprehensively organized. Every biomarker has been explained and the most recent evidence has been summarized properly. I suggest modifying the Tables that are difficult to follow and creating a new column for each endpoint. Are data available on the response to immunotherapy or adjuvant chemotherapy in those patients (PMID= 35200559, 39031261, 33128595)? This aspect is gaining more importance nowadays and should be acknowledged. 

Author Response

Comments 1: The authors explored the role of biomarkers (blood, tumor-derived, and urine-based) in upper tract urothelial carcinoma (UTUC). Specifically, they relied on a systematic review methodology including 94 studies. The work is clearly presented, well-read, and comprehensively organized. Every biomarker has been explained and the most recent evidence has been summarized properly.

Response: We would like to thank the Reviewer for his/her comments.

Comments 2: I suggest modifying the Tables that are difficult to follow and creating a new column for each endpoint.

Response: The tables have been altered. In each table we created three columns for the specific survival outcomes that were reported most commonly (OS, CSS, RFS). Also, a column with “other survival outcomes” was added, for the endpoints that were reported rarely. 

Comments 3: Are data available on the response to immunotherapy or adjuvant chemotherapy in those patients (PMID= 35200559, 39031261, 33128595)? This aspect is gaining more importance nowadays and should be acknowledged. 

Response: Thank you for this valuable suggestion. The vast majority of the prognostic data was based on patients undergoing radical nephroureterectomy/ureterectomy only. We strived to specify the treatment that each of the study groups received.

Unfortunately, few of the papers researched prognostic biomarkers in patients treated with neoadjuvant/adjuvant chemotherapy or immunotherapy. We changed the text accordingly, in order to highlight where these modalities were used:

-“Another study proved that pretreatment high CRP was a reliable prognostic factor of worse overall survival (OS) in clinical node-positive patients who underwent RNU, (neo)adjuvant chemotherapy, radiotherapy and/or palliative care [7].” (line 87); 

-“In addition, the latest study from 2024 reported that there was no correlation between serum creatinine and OS in 105 UTUC patients who underwent RNU, (neo)adjuvant chemotherapy, radiotherapy and/or palliative care [7].” (line 284);

-”A meta-analysis of 7 articles published in 2013 suggested that positive p53 expression was a potential prognostic marker in UTUC patients qualified to RNU and/or chemotherapy as it was strongly associated with DFS, CSS and OS [87].” (line 379);

-”However, Kovács et al., in the retrospective study, found that increased serum MMPs levels were significantly associated with worse OS and presence of lymph node or distant metastases in patients who were qualified for RNU, chemotherapy, immunotherapy or the combinations of these [48].” (line 314)

Nonetheless, we strongly agree that this is a very important topic, as such treatments gain popularity. Studies on this matter would be beneficial for the field.